# Synthesis of Electron-Rich Porous Organic Polymers via Schiff-Base Chemistry for Efficient Iodine Capture

**DOI:** 10.3390/molecules27165161

**Published:** 2022-08-12

**Authors:** Peng Tian, Zhiting Ai, Hui Hu, Ming Wang, Yaling Li, Xinpei Gao, Jiaying Qian, Xiaofang Su, Songtao Xiao, Huanjun Xu, Fei Lu, Yanan Gao

**Affiliations:** 1Key Laboratory of Ministry of Education for Advanced Materials in Tropical Island Resources, Department of Chemistry and Chemical Engineering, Hainan University, No 58, Renmin Avenue, Haikou 570228, China; 2China Institute of Atomic Energy, Beijing 102413, China; 3School of Science, Qiongtai Normal University, Haikou 571127, China

**Keywords:** porous organic polymers, iodine capture, radioiodine, electron-rich framework, charge-transfer complexes

## Abstract

As one of the main nuclear wastes generated in the process of nuclear fission, radioactive iodine has attracted worldwide attention due to its harm to public safety and environmental pollution. Therefore, it is of crucial importance to develop materials that can rapidly and efficiently capture radioactive iodine. Herein, we report the construction of three electron-rich porous organic polymers (POPs), denoted as POP-E, POP-T and POP-P via Schiff base polycondensations reactions between *Td*-symmetric adamantane knot and four-branched “linkage” molecules. We demonstrated that all the three POPs showed high iodine adsorption capability, among which the adsorption capacity of POP-T for iodine vapor reached up to 3.94 g·g^−1^ and the removal rate of iodine in n-hexane solution was up to 99%. The efficient iodine capture mechanism of the POP-T was investigated through systematic comparison of Fourier transform infrared spectroscopy (FT-IR), Raman spectroscopy and X-ray photoelectron spectroscopy (XPS) before and after iodine adsorption. The unique π-π conjugated system between imine bonds linked aromatic rings with iodine result in charge-transfer complexes, which explains the exceptional iodine capture capacity. Additionally, the introduction of heteroatoms into the framework would also enhance the iodine adsorption capability of POPs. Good retention behavior and recycling capacity were also observed for the POPs.

## 1. Introduction

Nuclear energy, as a clean energy source, has become one of the most realistic alternative energy sources in society today. The use of nuclear energy to generate electricity would greatly reduce carbon dioxide emissions, which will help alleviate the growing greenhouse effect of global warming. Nuclear energy is playing an increasingly important role in current energy supply due to its ultra-high energy density and low carbon emissions. However, the large amount of nuclear waste generated in the process of nuclear fission has seriously affected the environment and human health. So, how to deal with these nuclear wastes safely and efficiently is an urgent problem to be solved [1,2]. Radioactive iodine (^129^I and ^131^I) is one of the main nuclear wastes and is also considered to be a radioactive pollutant in nuclear power plants. ^129^I must be trapped and stored timely at disintegration because it has an exceptionally long half-live of 15.7 million years. Whereas ^131^I has a short lifetime of about 8 days, it needs to be immediately captured because of its highly mobile and volatile characteristics [3,4]. Therefore, it has become imperative to develop stable and effective methods for capture and storage of radioiodine.

To this end, various porous materials have been attempted for radioiodine capture. For example, inorganic porous materials such as activated carbon [5,6,7], zeolites [8,9,10] and silver-functionalized silica aerogels [11,12,13] have been used as adsorbents to capture radioiodine. However, these materials usually exhibit low absorption capacity due to their limited surface area and less interactive sites with iodine. Although metal-organic frameworks (MOFs) generally possess high surface area, their low absorption capacity and instability towards moisture has limited the practical applications [14,15,16,17]. In recent years, porous organic polymers (POPs) have also attracted a lot of attention for worthwhile iodine capture. POPs are a class of multi-dimensional porous organic materials, which are constructed via strong covalent linkages between various organic building units with different geometries and topologies. Depending on the degree of long-range order, POPs can be divided into amorphous materials, such as conjugated microporous polymers (CMPs) [18,19,20,21,22], porous aromatic frameworks (PAFs) [23,24,25,26], hyper-crosslinked polymers (HCPs) [27,28,29] and crystalline materials, including covalent triazine frameworks (CTFs) [30,31,32,33] and covalent organic frameworks (COFs) [34,35,36]. To date, a large number of COFs have been studied extensively for the capture of iodine and they have showed great potential as adsorbents for the effective removal of radioiodine [37,38,39]. However, it is known that the synthesis of crystalline COFs remains greatly challenging due to the lack of a universal protocol to guarantee high crystallinity of COFs. In many cases, a great deal of effort has to be expended on screening for optimal reaction conditions. Compared with crystalline COFs, amorphous POPs are much easier to be synthesized since POPs are generally formed via various C–C cross-coupling reactions, such as Suzuki [40,41], Heck [42], Sonogashira [43,44], Yamamoto [45], and Glaser [46] cross-coupling. The formation of C–C bonds are effectively catalysed by metal species and POPs are obtained with a high yield in large quantities. Besides, some POPs are linked to form through the condensation reactions of building units [47]. Therefore, amorphous POPs show better practical application prospects than crystalline COFs.

It is widely accepted that the electron donor character of the porous adsorbent is a critical parameter that will determine the I_2_ capture efficiency [48]. Therefore, the affinity and the accessibility of electron donor groups should be modulated when designing POPs framework architectures. Electron-rich structural units can be introduced into the framework, and the electron transfer ability could be enhanced by extending the conjugation of the porous POPs. In this way, it is more beneficial to improve the binding ability of iodine with POPs, so as to achieve the purpose of enhancing the iodine capture ability of POPs.

With these considerations in mind, we successfully synthesized a series of electron-rich POPs (POP-E, POP-T and POP-P), which were constructed through the linkage of imine bonds (Figure 1). The conjugated π-system and nitrogen atoms in the polymers enabled the strong interactions between the POPs and iodine [49]. Also, the Lewis acid-base interactions between the polar C–N bond and iodine molecules increased the uptake capacity of iodine [50]. In this study, POP-E, POP-T and POP-P showed good efficiency for volatile iodine capture, which could reach 3.49, 3.94 and 3.27 g·g^−1^, respectively. Moreover, the POP absorbents also exhibited great iodine removal ability in iodine-n-hexane solution; for instance, the removal efficiency for POP-T in 500 mg·L^−1^ iodine-n-hexane solution could reach up to 99%.

## 2. Results and Discussion

The synthesis route of POP-E, POP-T and POP-P was shown in Figure 1. The three POPs were constructed through the Schiff base condensation reaction in the presence of 1,2-dichlorobenzene and acetic acid at 120 °C. The synthesis details of the monomers and the corresponding ^1^H NMR results (Appendix A) are given in Appendix A. The successful formation of POPs (POP-E, POP-T and POP-P) was confirmed by Fourier transform infrared spectroscopy (FT-IR) using an attenuated total reflection (ATR) mode (Figure 1). For the FT-IR spectra of tetrakis (4-aminophenyl) ethene (TAPE), 4,4’,4’’,4’’’-([2,2’-bi(1,3-dithiolylidene)]-4,4’,5,5’-tetrayl)tetraaniline (TTF-4NH_2_) and 1,3,6,8-tetrakis (4-aminophenyl) pyrene (PyTTA) monomers, the absorption bands between 3400 and 3200 cm^−1^ are due to the N-H stretching vibration. The absorption peaks around 1700 cm^−1^ can be ascribed to the -C=O stretching vibration. After the Schiff-base condensation reaction, the absorption bands between 3400 and 3200 cm^−1^ almost disappeared in the FT-IR spectra of POP-E, POP-T and POP-P, indicating a high degree of polymerization between the amine and aldehyde groups of the monomers. Meanwhile, several new absorption peaks appeared after the reaction between the two units. The peaks at 1663 cm^−1^, 1624 cm^−1^ and 1623 cm^−1^ represented the imine bonds for POP-E, POP-T and POP-P, respectively. It is worth noting that there were still characteristic absorption peaks of the aldehyde group existing in the three POPs, which suggests incomplete reaction between the amine and aldehyde groups.

The amorphous structure of POP-E, POP-T and POP-P was confirmed by Powder X-ray diffraction (PXRD), as shown in Appendix A. Thermogravimetric analysis (TGA) was used to test the thermal stability of three POPs under nitrogen conditions and observed that the three POPs are thermal stable up to 200 °C. Furthermore, when the temperature reached 800 °C, there was still 54%, 35% and 45% of the weight retained by POP-E, POP-T and POP-P, respectively, indicating that all three POPs possess good thermal stability (Appendix A). The field-emission scanning electron microscope (FE-SEM) was used to observe the topographic features of the POPs (Appendix A). POP-E exhibited a honeycomb columnar structure. POP-T was mainly formed by disordered the stacking of several nano-spherical particles. POP-P was composed of disordered arrangement of granular solids with nanometric scale.

The nitrogen adsorption−desorption isotherm curves of POP-E, POP-T and POP-P were recorded at 77 K and shown in Figure 2. It can be seen that the nitrogen adsorption and desorption curves of these POPs were similar. When the relative pressure was lower than 0.1 (P/P_0_ < 0.1), the amount of nitrogen adsorption by the porous materials increased slowly, indicating that there existed microporous structures in the material. When the relative pressure was higher than 0.1 but lower than 0.9 (0.1 < P/P_0_ < 0.9), the adsorption capacity of the POPs for nitrogen increased slowly; there was a hysteresis loop between the nitrogen adsorption curve and the desorption curve of the material, suggesting the mesoporous nature of the three POP materials. Furthermore, when the relative pressure was in the range from 0.9 to 1.0 (0.9 < P/P_0_ < 1.0), the nitrogen adsorption by the materials increased sharply, indicating the co-existence of macropore structures in the three POPs. The test results showed that all three POPs possess hierarchical pore structures. The Brunauer–Emmett–Teller (BET) surface areas of POP-E, POP-T and POP-P were calculated as 37.0, 18.3 and 27.8 m^2^·g^−1^, respectively, while the corresponding total pore volume was estimated from single-point nitrogen uptake at P/P_0_ = 0.99 to be 0.027, 0.037 and 0.058 m^3^·g^−1^, respectively. Finally, the average pore size distribution of POP-E, POP-T and POP-P was calculated to be 1.4, 1.6 and 1.2 nm, respectively, according to nonlocal density functional theory pore size distribution (NLDFT), which indicated that all three POPs were dominated by microporous structures.

After understanding the structures and properties of the three POPs, the iodine vapor adsorption capacity of the POPs was explored. ^127^I was used because of its similar chemical properties to radioactive iodine nuclides [48]. A little amount of POP adsorbent and excess iodine were placed into an airtight system at 348 K under ambient pressure. The iodine uptake capacity of the samples was evaluated by calculating the weight changes of the samples before and after iodine vapor adsorption. (Details are given in Appendix A)

It can be seen from the iodine adsorption curve (Figure 3) that the iodine was rapidly adsorbed by POPs at 75 °C within the initial 20 h, which could be attributed to the hierarchical porous structure of the porous material [51]. Among the three POPs, POP-T showed the best iodine vapor uptake ability. After 30 h, the weight growth of the three POPs gradually slowed down, indicating that the adsorption of iodine vapor was close to saturation. Based on these observations, the iodine vapor adsorption experiments of the POPs were continued for another 42 h. It was found that when the adsorption time reached 72 h, the weight of POPs no longer changed obviously, indicating that the iodine uptake of POP-E, POP-T and POP-P was completely saturated. Meanwhile, the colour of the three POPs turned black after the saturated adsorption of iodine, which was related to the deposition of iodine on the surface of the POPs and suggested the high uptake degree of iodine. After calculation, the iodine adsorption capacity of POP-E, POP-T and POP-P was 3.49, 3.94 and 3.27 g·g^−1^, respectively, indicating that all three POPs have good iodine adsorption capacity.

Compared with reported POPs, it is obvious that POP-E, POP-T and POP-P exhibited outstanding iodine adsorption capacity, better than most other materials (Appendix A). Although the BET surface areas of the three POPs was not high, their iodine adsorption capacity was still strong, which shows that the specific surface area and pore volume of iodine adsorbents were not the decisive factors to determine the iodine adsorption performance of POPs. The high iodine capability of the POPs can be ascribed to the introduction of monomers with large π-π conjugated structural units into POPs that could expand the conjugation of the polymer network [37]. Among the three POPs, POP-T showed a better iodine capture performance. The introduction of heteroatoms (sulfer here) with lone pairs of electrons into the electron-rich building blocks could additionally increase the adsorption of iodine vapor [52]. Specifically, the microporous structure was beneficial for the adsorption of iodine, while the mesoporous and microporous structures were beneficial to the mass transfer process of iodine. These factors could endow POP-T with superior iodine adsorption performance.

To verify the mechanism of iodine adsorption, X-ray photoelectron spectroscopy (XPS) and Raman spectroscopy were used to detect the interaction between the captured iodine and the electron-rich framework of the POP-T. The POP with loaded iodine is referred to as I_2_@POP. In the XPS spectra (Appendix A), the N 1s of POP-T showed two binding energy peaks at 399.0 and 400.2 eV, which correspond to C=N and C-N, respectively. After iodine loading, the binding energy of 399.0 eV belonging to C=N slightly shifted to 399.1 eV, while a new peak with binding energy of 401.1 eV appeared. This new peak may have been shifted from 399 eV because of the interactions between lone pairs of electrons on nitrogen atoms and the uptake of electron-deficient iodine. In addition, comparing the Raman spectra of POP-T before and after adsorption of iodine, it could be clearly observed that new absorption peaks at 108 and 165 cm^−1^ were generated (Figure 4a), which could be assigned to the resonant modes of polyiodides, mainly I_3_^−^ and I_5_^−^ anions. The band at 108 cm^−1^ was ascribed to the symmetric and asymmetric stretching vibrations of I_3_^−^ and the peak at 165 cm^−1^ was assigned to the I_5_^−^ stretching vibration [53,54]. Moreover, the existence of polyiodide anions I_3_^−^ and I_5_^−^ was further confirmed by XPS spectra. Two groups of I 3d signals of iodine were observed in the XPS spectrum of I_2_@POP-T (Figure 4b). From the XPS spectrum, the I_3_^−^ signal could be observed at 618.2 and 629.8 eV, while the peaks at 619.6 and 631.1 eV were possibly due to the I_5_^−^ signal. As shown in the FTIR spectra (Appendix A), the peak at 1628 cm^−1^ ascribed to C=N blue-shifted to 1641 cm^−1^ while the peak at 1602 cm^−1^ related to C=C red-shifted to 1595 cm^−1^. Based on the previous reports [37,49] and the above results, it can be concluded that the π-π conjugated systems comprising of the phenyl rings and imine groups should be involved in the formation of charge transfer complex with iodine. As a result, the chemisorption via charge transfer led to the high iodine uptake capacity of POP-T.

As an important indicator to assess the performance of porous materials in nuclear industry, the iodine desorption capacity of porous materials is worth studying. Therefore, we investigated the iodine release behavior of I_2_@POP-T into methanol. The delivery of iodine from I_2_@POP-T into methanol was monitored by using UV/Vis spectroscopy (Appendix A). It was obvious that about 90% of adsorbed iodine was quickly released within 120 min, displaying the excellent desorption capacity of POP-T (Figure 5a). Also, the retention capacity of iodine captured in the pores of POP-T was also detected. TGA was used to study the retention capacity of I_2_@POP-T (Appendix A). The iodine was slowly removed at temperature higher than about 100 °C, suggesting a good retention behavior of POP-T. We can attribute this good retention capacity to strong chemisorption via charge transfer between the framework of the POP and captured iodine. Besides, it is expected that the absorbent can be cycled for repeated use. The iodine in pores of I_2_@POP-T was removed through Soxhlet extraction with methanol for 48 h. After being vacuum-dried at 120 °C for 6 h, samples are reused for iodine vapor absorption. The results showed that POP-T still possessed a high adsorption capacity of 3.2 g·g^−1^ after the fifth recycling, suggesting a good recyclability (Figure 5b).

In addition, we investigated the iodine adsorption capacity of POP-E, POP-T and POP-P in a solution environment. The adsorption experiment of iodine was carried out in n-hexane (details are given in Appendix A). The experimental results showed that all three POPs exhibited good iodine removal rates in n-hexane solution at different concentrations. Their iodine uptake capabilities were 199.7, 193.4 and 150.7 mg·g^−1^ for POP-E, POP-T and POP-P, respectively, when the concentration of iodine in n-hexane was 100 mg·L^−1^. The values were 546.1, 594.0 and 489.7 mg·g^−1^ and 610.5, 902.8 and 682.3 mg·L^−1^ when the concentration of iodine in n-hexane was 300 and 500 mg·L^−1^, respectively (Figure 6a). The removal efficiencies of POP-E, POP-T and POP-P were 99.9%, 96.7% and 75.3% in 100 mg·L^−1^ iodine-n-hexane solution: 91.0%, 99.0% and 81.6% in 300 mg·L^−1^ iodine-n-hexane solution and 61.1%, 90.3% and 68.2% in 500 mg·L^−1^ iodine-n-hexane solution, respectively (Figure 6b). Obviously, POP-T has the best iodine adsorption capacity in n-hexane among the three POPs, which is consistent with the adsorption performance of iodine vapor (Figure 6). The iodine adsorption curves of three POPs were determined in 300 mg· L^−1^ iodine-n-hexane solution monitored by UV/ Vis spectra (Appendix A). Due to the effect of solvent encapsulation, the adsorption capacity of POPs in iodine solution was lower than their adsorption capacity for iodine vapor.

## 3. Conclusions

In summary, three kinds of electron-rich porous organic polymers have been successfully constructed via a Schiff base reaction. The large π-π conjugated system between aromatic rings in the three POPs formed charge-transfer complexes with adsorbed iodine molecules, which afforded the POPs with strong iodine adsorption capacity. Although the three POPs feature low specific surface area and pore volume, they all showed good iodine adsorption capacity, which indicated that specific surface area and pore volume are not the prerequisite to determine the iodine adsorption performance of POPs. Furthermore, our results revealed that the iodine adsorption capacity of POPs can be enhanced by introducing electron-rich units into the framework of POPs, and this effect could be further enhanced by introducing heteroatoms into the electron-rich structure. The three POPs not only showed good iodine vapor adsorption capacity, but also exhibited great iodine removal performance in iodine-n-hexane solution. We envision that these results might open a novel way to design porous materials for efficient iodine as well as other pollutant capture.

## Data Availability

Not applicable.

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
