# Peer review of "Synthesis of Electron-Rich Porous Organic Polymers via Schiff-Base Chemistry for Efficient Iodine Capture"

_molecules, 2022, doi:10.3390/molecules27165161_

Round 1

Reviewer 1 Report

The manuscript by Peng Tian and collaborators with title “Synthesis of Electron-Rich Porous Organic Polymers Via Schiff-Base Chemistry for Efficient Iodine Capture” reported the construction of three electron-rich porous organic polymers (POPs) via Schiff base polycondensations reaction between Td-symmetric adamantane knot and four-branched "linkage" molecules. The studied POPs showed high iodine vapor adsorption capability up to 3.94 g·g1 and the removal rate of iodine in n-hexane solution was up to 99%. The efficient iodine capture mechanism was further investigated through systematic comparison of FT-IR, Raman spectroscopy and XPS before and after iodine adsorption. Based on current results and conclusion, I recommend publication of this paper in Molecules after addressing the comments. 

1.     Could the author comment on why select 1,3,5,7-tetrakis (4-formylphenyl) adamantane (TFPA) as the knot and TAPE, TTF-4NH2, PyTTA as the linker?  

2.     Figure S8 shows the PXRD spectra of POP-E, POP-T, and POP-P. The three POPs all exhibit kind of crystalline peaks. Have the author tried to improve the Schiff base polycondensations reaction to obtain COFs? And for the purpose of iodine capture, which kind of porous materials is more attractive?

3.     The author demonstrated the iodine capture ability of the POPs in vapor state and solution state. Is there any difference between the adsorption mechanism of the vapor adsorption and the solution adsorption? Furthermore, which one is more instructive for practical application of iodine capture? 

Reviewer 2 Report

In this manuscript, the authors reported the construction of three electron-rich porous organic polymers (POPs) via Schiff-Base polycondensation reaction between Td-symmetric adamantane knot and four-branched "linkage" molecules. All the three POPs showed high iodine adsorption capability. The large π-π conjugated system between imine bonds linked aromatic rings with adsorbed iodine result in the three charge-transfer complexes, which afforded the POPs with strong iodine adsorption capacity. This is an interesting work in design and syntheses of porous materials for efficient iodine as well as other adsorbents. But I did not find the file of supporting information, the authors needed to submit it to confirm their discussion and conclusions. Therefore, I recommend it to be published in Molecules after a major revision.

In details:

1. In this article, the authors said the morphology of the three POPs is characterized by SEM, did the author also do TEM characterizations.

2. In this article, only thermal stability tests are done for the three samples, did the authors check their chemical stability after immerging these samples in different solvents.

3. In the Abstract, it described that the adsorption mechanism was analyzed by IR, XPS and Raman spectroscopy, but the characterizations of IR before and after iodine adsorption is not covered in the main text.

Round 2

Reviewer 2 Report

The authors added the supporting materials now and answered the questions appropriately. It seems meet the requiements for publications now.